**REPORT**

# The molecular architecture of the meiotic spindle is remodeled during metaphase arrest in oocytes

Mariana F.A. Costa and Hiroyuki Ohkura

**Before fertilization, oocytes of most species undergo a long, natural arrest in metaphase. Before this, prometaphase I is also prolonged, due to late stable kinetochore–microtubule attachment. How oocytes stably maintain the dynamic spindle for hours during these periods is poorly understood. Here we report that the bipolar spindle changes its molecular architecture during the long prometaphase/metaphase I in *Drosophila melanogaster* oocytes. By generating transgenic flies expressing GFP-tagged spindle proteins, we found that 14 of 25 spindle proteins change their distribution in the bipolar spindle. Among them, microtubule cross-linking kinesins, MKlp1/Pavarotti and kinesin-5/Klp61F, accumulate to the spindle equator in late metaphase. We found that the late equator accumulation of MKlp1/Pavarotti is regulated by a mechanism distinct from that in mitosis. While MKlp1/Pavarotti contributes to the control of spindle length, kinesin-5/Klp61F is crucial for maintaining a bipolar spindle during metaphase I arrest. Our study provides novel insight into how oocytes maintain a bipolar spindle during metaphase arrest.**

## Introduction

A unique feature of oocytes is that centrosomes are eliminated during oogenesis in many animals (Szollosi et al., 1972; McKim and Hawley, 1995; Holubcová et al., 2015). Centrosomes dictate bipolar spindle formation in mitosis by acting as the major microtubule nucleators (Kirschner and Mitchison, 1986). In oocytes, without centrosomes, a functional spindle is formed and maintained through cooperation of multiple microtubule motors, cross-linkers, and regulators of microtubule dynamicity (Heald et al., 1996; Walczak et al., 1998; Ohkura, 2015).

Another unique feature of oocytes is that they naturally arrest the cell cycle twice: in late meiotic prophase I and metaphase I/II (Bennett, 1977). The second arrest can last for many hours before fertilization (12–24 h in humans and a few hours to a few days in *Drosophila melanogaster*; David and Merle, 1968; King, 1970; Spence and Mason, 1992; Hughes et al., 2018) and takes place in metaphase I or II depending on the species (Sagata, 1996). Furthermore, oocytes commonly delay stable microtubule attachment to kinetochores, resulting in a long prometaphase I (Brunet et al., 1999; Kitajima et al., 2011; Głuszek et al., 2015).

Although microtubules turn over rapidly within a bipolar spindle, the spindle as a whole maintains its overall shape and polarity (Mitchison and Kirschner, 1984; Colombié et al., 2013). In contrast to mitosis, which takes place relatively quickly, accurate chromosome segregation in oocytes requires that a bipolar spindle is maintained during the long prometaphase I and during metaphase arrest. Limited information is available about how the dynamic spindle is stably maintained during these periods (Lefebvre et al., 2002; Terret et al., 2003).

We hypothesized that localization changes of crucial microtubule regulators may stabilize the bipolar spindle during the long prometaphase I or metaphase arrest in oocytes. Here we report that a substantial number of spindle proteins change their localization during these periods in *Drosophila* oocytes. Crucially, two microtubule cross-linking kinesins, MKlp1/Pavarotti (Pav) and kinesin-5/Klp61F, accumulate in the spindle equator only later during the metaphase I arrest. An oocyte-specific mechanism regulates MKlp1/Pav, which contributes to spindle length control. Importantly, kinesin-5/Klp61F is crucial for maintaining the bipolar spindle during metaphase I arrest.

## Results and discussion

### Identifying localization changes of spindle proteins during prometaphase/metaphase I in oocytes

Oocytes commonly spend a long time in meiotic prometaphase I (Brunet et al., 1999; Kitajima et al., 2011; Głuszek et al., 2015) and also naturally arrest in metaphase I or II before fertilization in most species (Sagata, 1996). We hypothesized that the composition or distribution of microtubule regulators within the bipolar spindle changes to confer extra spindle stability during the arrest.

Wellcome Centre for Cell Biology, School of Biological Sciences, University of Edinburgh, Edinburgh, UK.

Correspondence to Hiroyuki Ohkura: h.ohkura@ed.ac.uk.

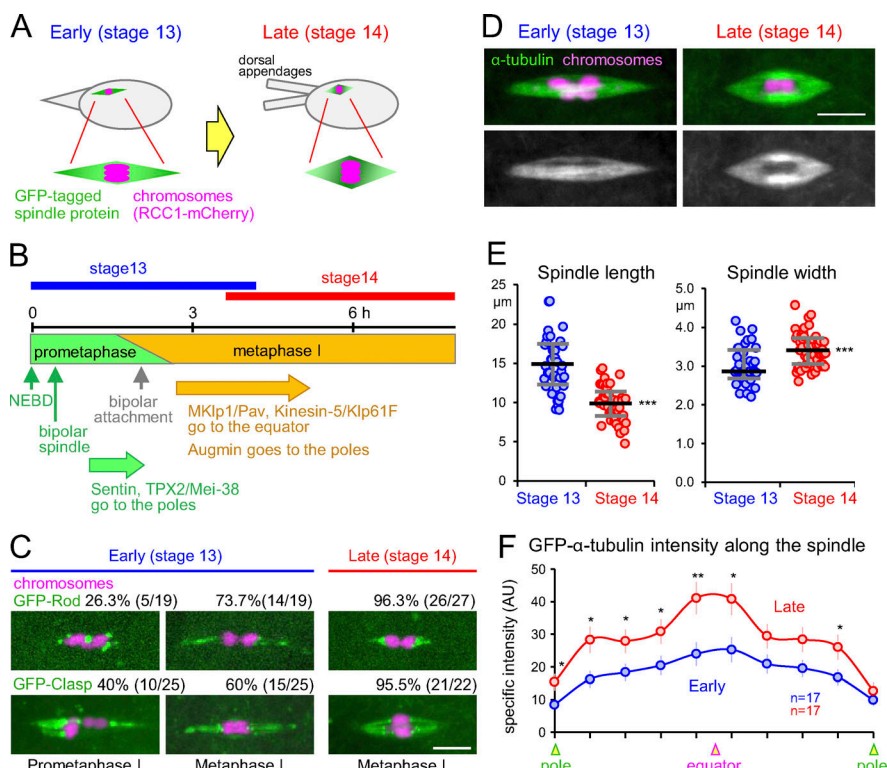

Figure 1. **The spindle architecture changes during meiotic arrest.** (A) Diagram comparing the localization of spindle proteins between early (stage 13) and late (stage 14) oocytes. (B) The estimated timings of events from snapshot and time-lapse imaging. (C) Representative prometaphase I and metaphase I figures in oocytes expressing GFP-Rod or Clasp and Rcc1-mCherry, together with the frequencies. Bar = 5 μm. (D) Change of the spindle architecture during the arrest observed in oocytes expressing GFP-α-tubulin and Rcc1-mCherry. Bar = 5 μm. (E) The spindle is longer and wider in stage 14 than stage 13. (F) GFP-α-tubulin intensity along the spindle. After maximum-intensity projection on the spindle axis for each spindle, the average specific signal intensity in arbitrary units above the background signal was plotted for each of 10 equally divided segments along the axis. Error bars indicate SEM. *, $P < 0.05$; **, $P < 0.01$; ***, $P < 0.001$. AU, arbitrary units.

To test our hypothesis, we used *Drosophila* oocytes, which undergo prolonged prometaphase I and naturally arrest in metaphase I before ovulation/fertilization. The localization of crucial microtubule regulators was compared between early and late stages after establishing spindle bipolarity (Fig. 1 A). We systematically generated transgenic flies expressing GFP-tagged proteins (Fig. S1) and then introduced Rcc1-mCherry to mark chromosomes. Oocytes were dissected from females matured in the presence of males and food, which allows ovulation/fertilization. The fluorescent signals were captured in live oocytes ("snapshots"). The morphology of dorsal appendages of oocytes (King, 1970; Gilliland et al., 2009) was used to distinguish between the last stages of oogenesis (stage 13 and 14). In this report, we refer to stage 13 and 14 oocytes as "early" and "late" oocytes, respectively, and compared the localization between these two stages.

### Spindle poles and equator are already established early in oocytes

Among 25 spindle proteins examined, 11 showed no or little change in localization between stage 13 and 14 (Fig. S1, A and C). Of these, the kinesin-6 MKlp2/Subito, Cyclin B, and a subunit of the chromosomal passenger complex, Incenp, were concentrated to the spindle equator in both stages (Fig. 2 A). In contrast, the kinesin-13 MCAK/Klp10A and the microtubule-associated protein Hurp/Mars were concentrated toward the spindle poles in both stages (Fig. 2 B). This indicates that the spindle equator and poles are established early, and that the localization changes of other equator or pole proteins described below do not simply reflect passive consequences in the underlying bipolar spindle microtubule organization.

Proteins that changed localization between stage 13 and 14 include kinetochore proteins whose behavior changes in response to microtubule attachment (Fig. 1 C and Fig. S1, A and B). Among them, Rod and Clasp are known to translocate along kinetochore microtubules upon attachment (Basto et al., 2004; Reis et al., 2009; Głuszek et al., 2015). Using these markers and chromosome configuration, we classified oocytes into prometaphase I and metaphase I (Fig. 1 C). In metaphase I, chromosomes are aligned symmetrically, and Rod and Clasp localize to kinetochore microtubules extending symmetrically to both poles. In prometaphase I, Rod/Clasp are concentrated solely on kinetochores (unattached), or chromosomes and kinetochore microtubules highlighted by Rod/Clasp are asymmetrically arranged (monopolar attachment). We found that stage 13 oocytes were a mixture of prometaphase I and metaphase I, and nearly all stage 14 oocytes were in metaphase I (Fig. 1 C).

### The microtubule architecture of the bipolar spindle changes during prometaphase/metaphase I

It was previously suggested that the spindle becomes shorter during the arrest (Gilliland et al., 2009). To test this, we compared stage 13 and 14 oocytes expressing GFP-α-tubulin and Rcc1-mCherry (Fig. 1 D). The spindles were shorter and wider in stage 14 than stage 13 (Fig. 1 E).

Interestingly, spindles in stage 14 oocytes had a higher intensity of α-tubulin, especially in the equator region where antiparallel microtubule arrays are thought to predominate (Fig. 1 F). Therefore, the microtubule architecture of the spindle had changed during prometaphase/metaphase I. A shorter and fatter spindle with a higher microtubule density in the equator in stage 14 oocytes may contribute to stabilizing spindle bipolarity.

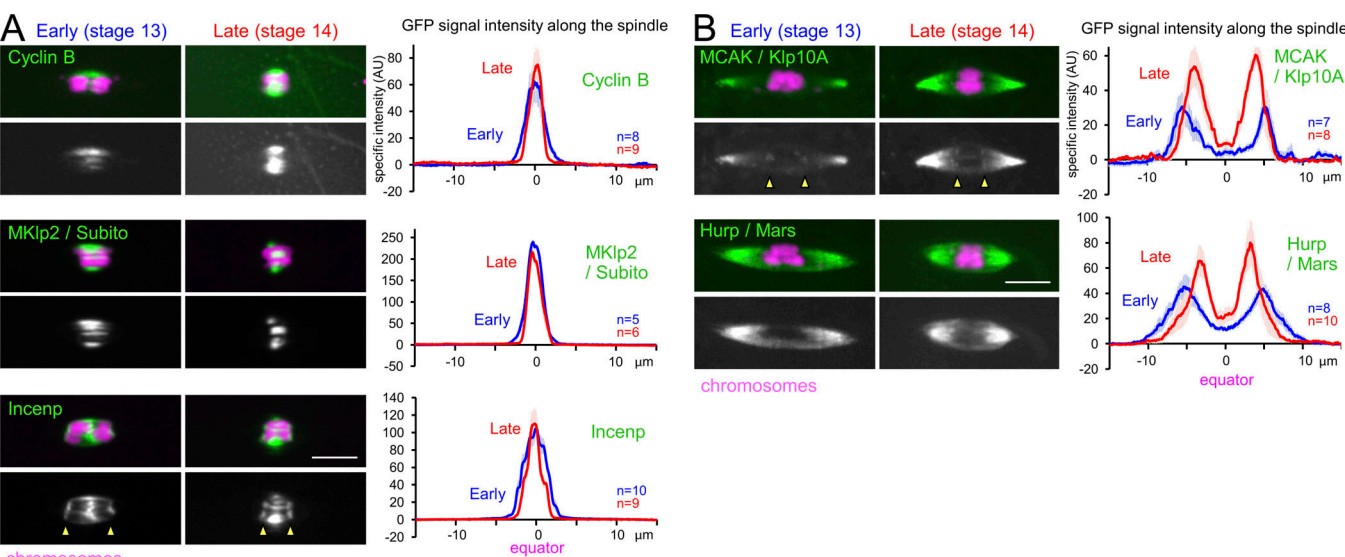

Figure 2. **Proteins constitutively concentrated in the spindle equator or at the poles in both early and late oocytes. (A and B)** Proteins constitutively concentrated in the spindle equator (A) or poles (B) in both early and late oocytes. Left: GFP-tagged proteins and chromosomes (Rcc1-mCherry) in stage 13 and 14 oocytes expressing these proteins. Right: The GFP signal intensity along the spindle. After maximum-intensity projection on the spindle axis for each spindle, the average specific signal intensity above the background signal was plotted with SEM along the axis. Arrowheads indicate presumed kinetochore localization. Bar = 5 µm. AU, arbitrary units.

## Many spindle proteins change localization pattern between early and late oocytes

We found that 9 of the 20 nonkinetochore spindle proteins examined drastically changed their localization between stage 13 and 14 (Fig. S1). Crucially, these proteins accumulated to the spindle equator or poles more strongly in stage 14 than stage 13 (Fig. 3). Late accumulation of these proteins may contribute to stabilizing spindle bipolarity in late oocytes.

### Proteins that accumulate to the spindle equator only in late oocytes

One of two kinesin-6s, MKlp1/Pav, and the microtubule cross-linker PRC1/Feo showed no or weak localization throughout the spindle in stage 13. In contrast, they strongly accumulated to the spindle equator in stage 14 (Fig. 3 A). Kinesin-5/Klp61F was initially concentrated toward the spindle poles in stage 13, while it was strongly concentrated in the spindle equator in stage 14 (Fig. 3 A). Microtubule cross-linking activity of these three proteins (Kapitein et al., 2008; van den Wildenberg et al., 2008; Davies et al., 2015) may stabilize the spindle through equator accumulation in late oocytes.

### Proteins that accumulate to the spindle poles in late oocytes

A subunit of the microtubule nucleating complex Augmin, Dgt6, and the microtubule-associated protein TPX2/Mei-38 were concentrated to the spindle poles in both stage 13 and 14, but much more strongly in stage 14 (Fig. 3 B). As we previously reported (Głuszek et al., 2015), the microtubule plus end protein Sentin localized throughout the spindle in stage 13 but was concentrated toward the poles in stage 14 (Fig. 3 B). The microtubule polymerase XMAP215/Msps and its associated protein TACC were concentrated toward the poles in both stage 13 and 14. In addition, they localized as foci on the

spindle more frequently in stage 13 than stage 14 (Fig. 3, B and C). Pole accumulation of these promoters of microtubule polymerization may also enhance the stability of spindle bipolarity in late oocytes.

## Two kinesins accumulate to the spindle equator in late metaphase

Differences of protein localization between stage 13 and 14 may represent changes during prometaphase I or during metaphase I arrest. To distinguish them, we roughly estimated the timing of changes by a small number of time-lapse imaging from nuclear envelope breakdown (NEBD; Fig. S2).

To estimate the timing of metaphase I onset (defined by achieving bipolar attachment of kinetochores), GFP-Clasp and Rod-GFP were followed together with Rcc1-mCherry (Fig. S2). Roughly 2 h after NEBD, both chromosomes and kinetochore microtubules highlighted by Rod/Clasp became symmetrically arranged (bipolar attachment). Therefore, we estimated that metaphase I starts roughly 2 h after NEBD. We further estimated that metaphase I in flies lasts ∼6 h or longer at 21°C under our experimental conditions (see Materials and methods).

A constitutive equator protein, GFP-Subito, started accumulating to the spindle equator several minutes after NEBD. GFP-Sentin and GFP-TPX2/Mei-38 started accumulating to the spindle poles ∼1–2 h after NEBD, which corresponds to prometaphase I. In contrast, GFP-MKlp1/Pav and GFP-kinesin-5/Klp61F started accumulating to the spindle equator ∼3 h after and fully accumulated ∼5 h after NEBD. With similar timing, the Augmin subunit Dgt6 has increased pole accumulation. Therefore, the important microtubule regulators MKlp1/Pav, kinesin-5/Klp61F, and Augmin change localization during metaphase I arrest.

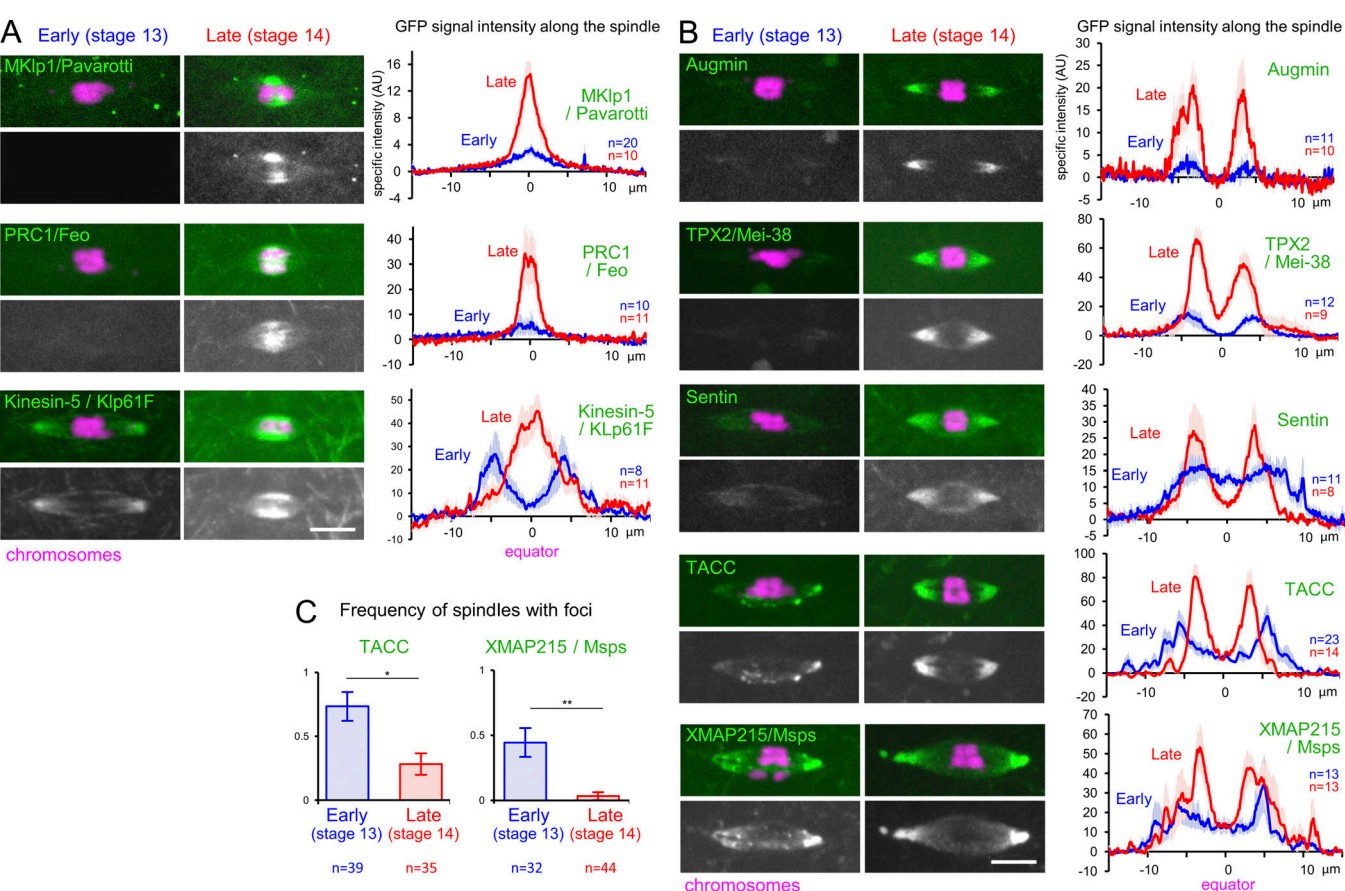

**Figure 3. Proteins that accumulate to the spindle equator or poles in late oocytes. (A and B)** Proteins accumulate to the spindle equator (A) or poles (B) in late oocytes. The data are presented as in Fig. 2. **(C)** The frequency of spindles with Msps or TACC foci; mean ± SEM. *, P < 0.05; **, P < 0.01. AU, arbitrary units.

## A mechanism distinct from mitosis regulates MKlp1/Pav accumulation in late oocytes

We then focused our further studies on two microtubule cross-linking kinesins, MKlp1/Pav and kinesin-5/Klp61F, which accumulate to the spindle equator in late metaphase. We first confirmed that the MKlp1/Pav localization was not caused by altered expression levels and that GFP-tagging did not disrupt the function (Fig. S3, A–C).

In mitosis, removal of inhibitory Cdk1 phosphorylation from MKlp1/Pav triggers its accumulation to the spindle equator in anaphase (Mishima et al., 2004; Goshima and Vale, 2005). In contrast, in *Drosophila* oocytes, MKlp1/Pav accumulates to the spindle equator in late metaphase (Figs. 3 A and S2). One hypothesis would be that, in oocytes, a pathway equivalent to mitotic anaphase is activated during meiotic metaphase arrest.

In *Drosophila* cultured cells, nonphosphorylatable mutations at four putative Cdk1 phosphorylation sites of MKlp1/Pav prematurely localize it to the spindle equator in mitotic metaphase (Goshima and Vale, 2005). To test whether these sites also regulate the MKlp1/Pav localization in embryos and oocytes, we made a nonphosphorylatable version (MKlp1/Pav-4A) and a phospho-mimetic version (MKlp1/Pav-4D). The endogenous

MKlp1/Pav protein was depleted by expressing shRNA together with an RNAi-resistant form of each GFP-MKlp1/Pav variant (Fig. S3 D).

In mitotic embryos, wild-type GFP-MKlp1/Pav accumulated in the spindle equator/central spindle only after late anaphase/telophase (Fig. 4, A and C), as reported for the endogenous MKlp1/Pav (Adams et al., 1998). In contrast, GFP-MKlp1/Pav-4A prematurely associated with the spindle in metaphase, while accumulation of GFP-MKlp1/Pav-4D was greatly reduced in late anaphase/telophase (Fig. 4, A and C). Therefore, these phosphorylations inhibit premature accumulation of MKlp1/Pav to the spindle until late anaphase in mitosis.

In oocytes, GFP-MKlp1/Pav accumulated to the equator of the metaphase spindle only in stage 14 (Fig. 4, B and D). GFP-MKlp1/Pav-4A prematurely accumulated to the spindle equator in stage 13, and more strongly in stage 14. Remarkably, GFP-MKlp1/Pav-4D accumulated to the spindle in stage 14 as strongly as wild-type MKlp1/Pav (Fig. 4, B and D). These results showed that MKlp1/Pav association to the early metaphase spindle in oocytes is prevented by the same mechanism as in mitotic metaphase. However, the equator accumulation in late metaphase in oocytes is regulated by a mechanism distinct from that in mitosis.

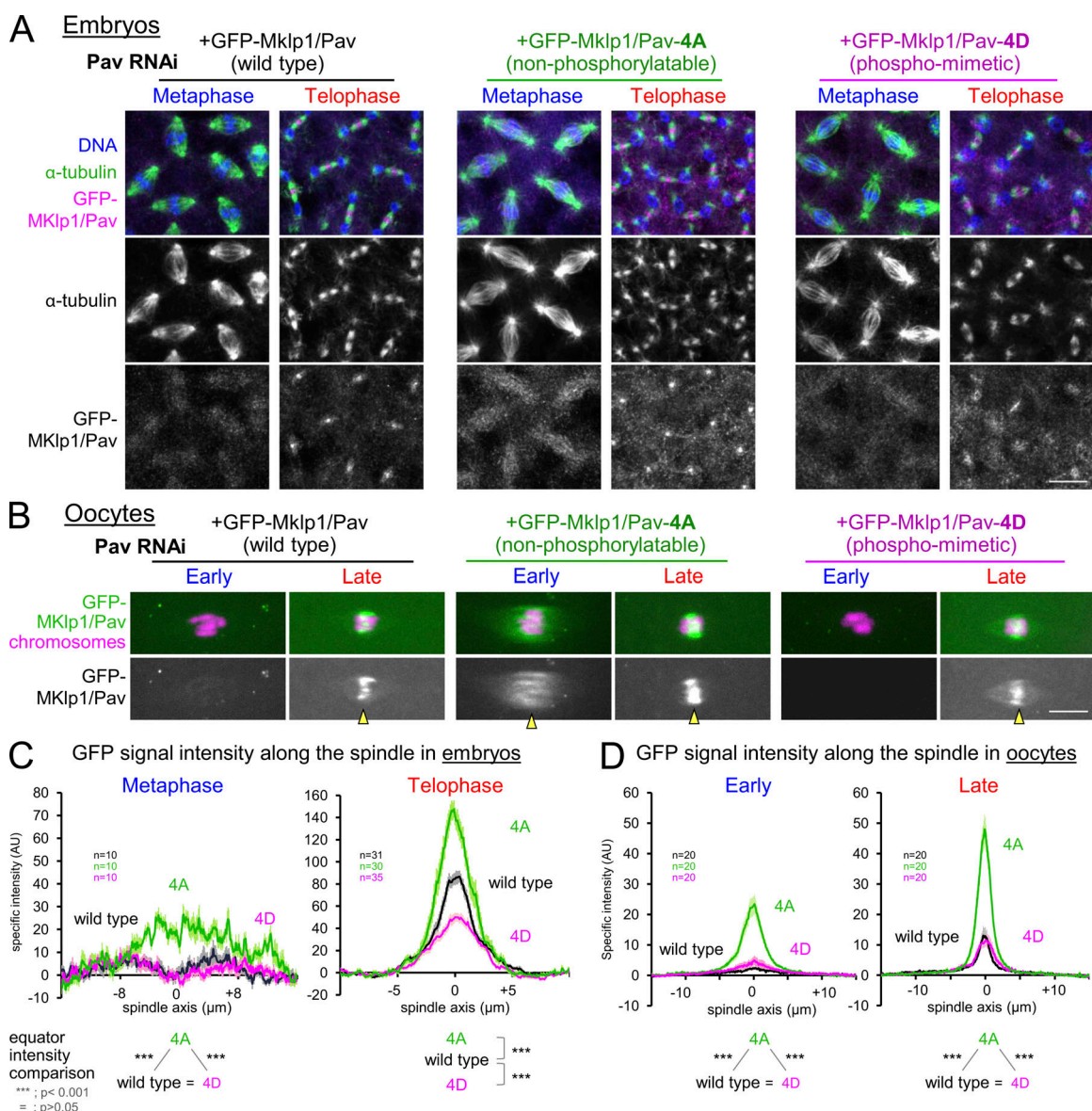

Figure 4. **A mechanism distinct from mitosis regulates the equator accumulation of MKlp1/Pav in late oocytes. (A)** Localization of GFP-tagged MKlp1/Pav phospho mutants in mitotic metaphase and telophase in syncytial embryos depleted of the endogenous MKlp1/Pav. Bar = 10 µm. **(B)** Localization of GFP-tagged MKlp1/Pav phospho mutants in stage 13 (early) and stage 14 (late) oocytes depleted of the endogenous MKlp1/Pav. The arrowheads indicate accumulation of MKlp1/Pav in the spindle equator. Bar = 5 µm. **(C and D)** The intensities of GFP-tagged MKlp1/Pav signal along the spindle axis in embryos and oocytes, quantified as in Fig. 2. ***, P < 0.001.

## Kinesin-5/Klp61F is important for maintaining a bipolar spindle during metaphase I arrest

In mitosis, MKlp1/Pav does not localize to or have a function for the metaphase spindle (Mishima et al., 2004; Glotzer, 2009). To test its function in oocytes, we depleted MKlp1/Pav using a strong driver of shRNA, which prevented formation of mature oocytes due to its role in early oogenesis (Minestrini et al., 2002). Immunostaining of predominantly stage 14 oocytes showed that a partial depletion of MKlp1/Pav using a weaker driver (Fig. S3 E) resulted in significantly longer and narrower spindles in oocytes than control (Fig. 5, A and B). This spindle morphology resembles that of stage 13 wild-type oocytes (Fig. 1 C), in which MKlp1/Pav does not yet accumulate to the spindle

equator (Fig. 3 A). Although this is a mild effect, these data are consistent with the possibility that MKlp1/Pav accumulation to the spindle equator contributes to shortening and widening of the spindle.

It was previously reported that immunostaining of oocytes depleted of kinesin-5/Klp61F showed a mixture of bipolar spindles and disintegrated spindles (Radford et al., 2017). Kinesin-5/Klp61F localization changes during metaphase arrest from the poles to the equator, and these two localizations may have distinct functions in different stages. To test this possibility, we depleted kinesin-5/Klp61F and observed spindle morphology in live stage 13 and 14 oocytes expressing GFP-α-tubulin. Strikingly, while nearly all stage 13 oocytes had a bipolar spindle, the

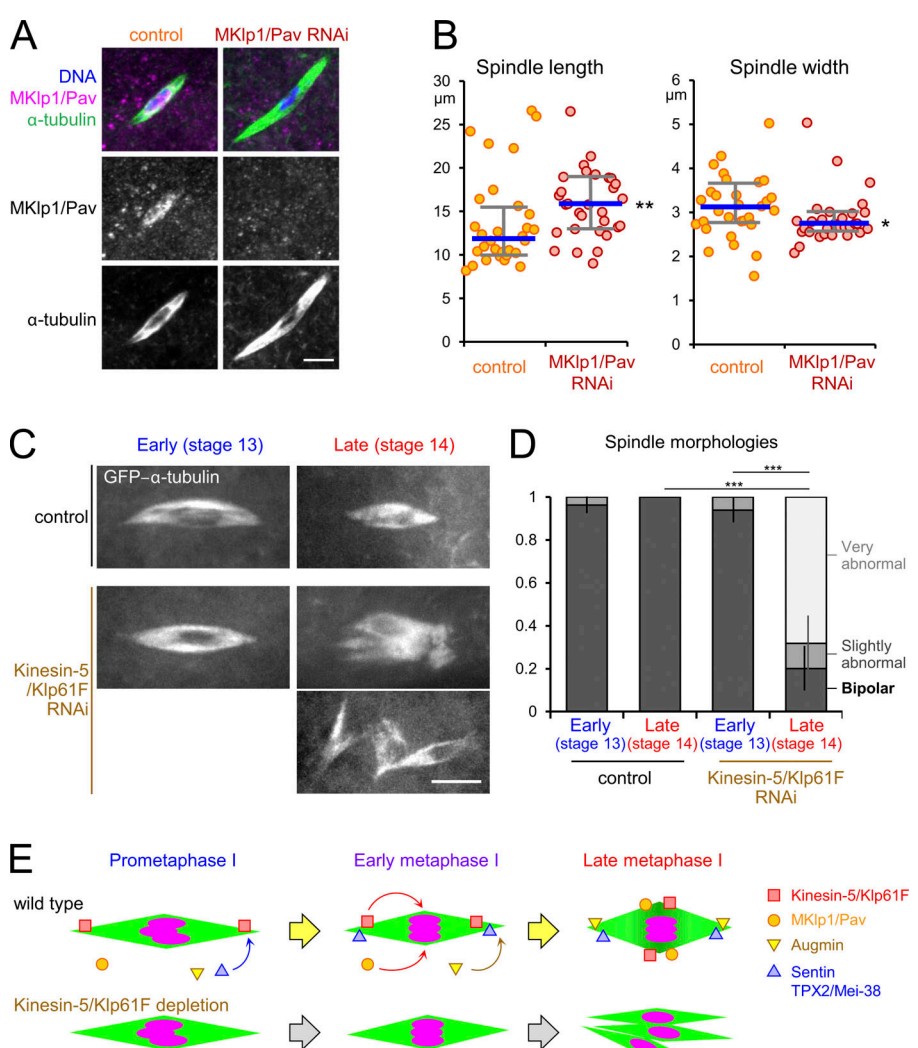

Figure 5. **MKlp1/Pav and kinesin-5/Klp61F contribute to the control of spindle length and bipolar spindle maintenance, respectively.** **(A)** Spindles in predominantly stage 14 oocytes with (MKlp1/Pav RNAi) or without (control) expressing shRNA against *MKlp1/Pav*. Bar = 5 µm. **(B)** The length and width of the spindles in control and MKlp1/Pav RNAi oocytes. Thick lines and error bars indicate the medians and the first/third quartiles, respectively. **(C)** Spindles in stage 13 and 14 oocytes expressing GFP-α-tubulin and depleted of kinesin-5/Klp61F by RNAi, and control without depletion. Bar = 5 µm. **(D)** The mean frequencies of spindle morphologies with SEM in the control and kinesin-5/Klp61F RNAi from three batches making up a total of 21, 30, 23, and 44 oocytes each. **(E)** A schematic model summarizing the change of the molecular architecture of the spindle in oocytes. *, $P < 0.05$; **, $P < 0.01$; ***, $P < 0.001$.

majority of the spindles in stage 14 oocytes were severely disrupted (Fig. 5, C and D). Expression of GFP-α-tubulin itself did not significantly disrupt spindle bipolarity (Fig. 5, C and D). Stage 13 and 14 roughly correspond to prometaphase/early metaphase I and late metaphase I, respectively. Therefore, kinesin-5/KLP61F is required for maintaining the bipolar spindle during metaphase arrest when it localizes to the equator, but is less important, if at all, in earlier stages when it localizes to the poles. This suggests that stage-specific localization of microtubule regulators is biologically important for accurate meiosis in oocytes.

### The molecular architecture of the spindle is remodeled during metaphase arrest in oocytes

In this study, we discovered a previously unknown phenomenon that provides a vital clue to a fundamental yet poorly understood question: how can oocytes stably maintain the bipolar spindle without centrosomes for a long time? Oocytes commonly spend a long time in prometaphase I and also naturally arrest in meiotic metaphase before fertilization. During these periods, the bipolar spindle must be stably maintained to segregate chromosomes accurately. In *Drosophila*, oocytes can naturally arrest in metaphase I up to a few days in different environmental conditions (King, 1970).

Our systematic analysis in *Drosophila* oocytes showed that many spindle proteins change their localization after establishing the spindle bipolarity (Fig. 5 E). In other words, the molecular architecture of the spindle is dramatically remodeled during the long prometaphase and metaphase I. We found that multiple microtubule regulators accumulate to the spindle poles or equators only in late oocytes, which could collectively stabilize the bipolar spindle. Crucially, two microtubule cross-linking kinesins, MKlp1/Pav and kinesin-5/Klp61F, accumulate to the spindle equator only in late metaphase. This late equator accumulation of MKlp1/Pav is achieved by an uncharacterized mechanism distinct from that in mitosis. Our results suggest that MKlp1/Pav contributes to the shortening and widening of the spindle in oocytes (Fig. 5, A and B). On the other hand, kinesin-5/Klp61F is crucial to maintain the bipolar spindle during metaphase I arrest, but less important, if at all, in earlier stages (Fig. 5, C and D). This supports our hypothesis that the change of kinesin-5/Klp61F localization to the equator during the arrest is critical for the stability of arrested spindles. Therefore, our results reveal the importance of the dynamic relocalization of spindle proteins during the arrest.

Vertebrate oocytes also undergo a long prometaphase I and metaphase II arrest. It is possible that equivalent spindle remodeling may occur in vertebrate oocytes. In addition, some microtubule regulators may be expressed or localized to the spindle only in the second division, with some examples being previously described (Lefebvre et al., 2002). Therefore, expanding this study into vertebrate oocytes may reveal a general principle on how oocytes maintain spindle bipolarity during arrest.

## Materials and methods

### Molecular techniques
Expression plasmids were generated by the use of the Gateway cloning system (Invitrogen). To generate entry plasmids, open reading frames were first PCR amplified from cDNAs using PrimeStar polymerase (Takara). They were inserted into the pENTR vector using the pENTR Directional TOPO Cloning kit (Invitrogen) or by Gibson assembly (New England Biolabs) after pENTR was linearized with NotI and AscI. The following cDNAs were used as templates for PCR: FIO3252 (MKlp1/Pav), LD23613 (Cyclin B), LD21642 (Hurp/Mars), LD31673 (CLASP), LD44443 (Cyclin A), LD21815 (Klp3A), LD14121 (Dgt6/Augmin), RE52507 (Incenp), LD35138 (Subito), and LD11851 (Polo). The cDNAs were provided by the Drosophila Genomics Resource Center. For Asp and Klp61F, genomic DNA from $w^{1118}$ flies was used, and for Mei-38, a cDNA library was used as a PCR template.

MKlp1/Pav mutations 4A (T7A/T15A/T458A/T467A) and 4D (T7D/T15D/T458D/T467D) were introduced by PCR-amplifying a pENTR vector carrying the MKlp1/Pav open reading frame using overlapping mutation-bearing primers and recircularizing using Gibson assembly (New England Biolabs). To avoid the transgenes being recognized by shRNA, we introduced four silent mutations (underlined; CACACCGTGCTGAACGCAGAA) into the part of the Mklp1/Pav transgene sequence that is recognized by the shRNA. We introduced these mutations using Gibson assembly, as described above for 4A and 4D mutations. Mutated plasmids were sequenced to confirm the absence of undesired mutations.

The different inserts in the pENTR vector were transferred to the selected destination vector by the LR recombination reaction using LR Clonase II enzyme (Invitrogen). Destination vectors pPGW and pPWG from the Drosophila Gateway vector collection made by T. Murphy (Carnegie Institution of Washington, Washington, DC) were used. The φPGW destination vector (upstream activator sequence p [UASp] promoter) was modified from the pPGW Gateway vector, into which attB had been inserted at the AatII site for PhiC31 integrase–mediated transgenesis. The φMaGW destination vector was modified from the φPGW destination vector in which the UASp promotor was replaced by the maternal α-tubulin 67C promoter.

### *Drosophila* genetics and techniques
Standard fly techniques were followed during this project according to Ashburner et al. (2005). Control oocytes used in this study were from $w^{1118}$ or $w^{1118}$ flies crossed with a specific GAL4 driver line. GAL4 driver lines used were V2H (P[Matα-Tubulin67C-Gal4]V2H; Bloomington Drosophila Stock Center [BDSC] 7062), V37 (P[Matα-Tubulin67C-Gal4]V37; BDSC 7063), and MVD1 (P[GAL4::VP16-nos.UTR]CG6325$^{MVD1}$; BDSC 4937). UASp-GFP-α-Tub84B (BDSC 7373) recombined with V37 or MVD1 was used for live imaging of the meiotic spindle in Figs. 1, 5 D, and S2 A and showed the same results.

For live imaging of kinetochores and kinetochore microtubules in Figs. 1 C and S2 A, we used GFP-Rough deal (Rod) controlled by its own promoter (a gift from R. Karess, Institut Jacques Monod, Paris, France; Basto et al., 2004; Głuszek et al., 2015) and GFP-Clasp controlled by the maternal α-tubulin at 67C promoter (mat-α-tub67C-GFP-CLASP, this study).

For live imaging of the meiotic chromosomes, Rcc1-mCherry under the maternal α-tubulin at 67C promoter (Figs. 1 and 4B) or the UASp promoter (Figs. 2 and 3) was recombined with MVD1 (Figs. 1, 2, and 3) or V37 (Fig. 4 B) on the third chromosome. For live imaging of GFP-tagged spindle proteins in Figs. 2, 3, and S2, the following lines were crossed with a line carrying Rcc1-mCherry MVD1/V37: mat-α-tub67C-GFP-Pavarotti (this study), Ubi-p63E-Feo-GFP (BDSC 59274), UASp-GFP-Klp61F (this study), UASp-GFP-Dgt6/Augmin (this study), UASp-GFP-Mei-38 (this study), UASp-GFP-Sentin (Głuszek et al., 2015), native promoter-Msps-GFP (Brittle and Ohkura, 2005), Ubi-TACC-GFP (BDSC 7066), mat-α-tub67C-GFP-Cyclin B (this study), UASp-GFP-Subito (Romé and Ohkura, 2018), UASp-Incenp-GFP (this study), Ubi-p63E-Klp10A-GFP (BDSC 55127), and mat-α-tub67C-GFP-HURP/Mars (this study).

For live imaging of GFP-tagged spindle proteins in Fig. S1 (B and C), the following lines were used: UASp-Polo-GFP (this study), native promoter-Mad2-GFP (BDSC 35820), mat-α-tub67C-GFP-Cyclin A (this study), mat-α-tub67C-GFP-Asp (this study), Ubi-EB1-GFP (Shimada et al., 2006), Ubi-Klp67A-GFP (BDSC 35511), Ubi-p63E-Patronin-GFP (BDSC 55129), and mat-α-tub67C-GFP-Klp3A (this study).

In Fig. S3 (A and B), a fly line expressing endogenous GFP-MKlp1/Pav under its native promoter (CRISPR) was generated by InDroso Functional Genomics (Rennes, France). In Fig. S3 C, the UASp-GFP-MKlp1/Pav flies (Minestrini et al., 2002) were a gift from D. Glover (University of Cambridge, Cambridge, UK). The shRNA fly lines for Pav (HMJ02232; BSDC 42573) and KLp61F (HMS00552; BSDC 33685) used in Fig. 5 were produced by the Transgenic RNAi Project at Harvard Medical School (Cambridge, MA).

To generate transgenic fly lines expressing GFP-tagged spindle proteins in Figs. 2 and 3, phiC31 integrase–mediated transgenesis onto the third chromosome was performed by BestGene using the VK33 site (BDSC 9750). To generate transgenic flies for the various GFP-MKlp1/Pav mutants in Fig. 4, the VK37 site on the second chromosome (BDSC 9752) was used.

### Live imaging of the spindle in oocytes
For live imaging of *Drosophila* oocytes, <1-d-old adult females were matured in the presence of males and their food was supplemented with dried yeast for 3–7 d at 18°C or 3–5 d at 25°C. This allows natural ovulation and fertilization. The ovaries were dissected at room temperature in a drop of Halocarbon 700 oil (Halocarbon) on a coverslip (24 × 50 mm; Głuszek et al., 2015).

We roughly estimated that metaphase I lasts ≥6 h at 21°C under our experimental conditions. This estimate is based on the following observations. First, mature female flies carry more stage 14 oocytes than stage 13 oocytes. Second, immediately after dissection, 26–40% of spindles in stage 13 oocytes are in prometaphase I. Finally, in our time-lapse imaging at 21°C, prometaphase I lasted ~2 h, with bipolar spindles present for 1.5 h. It is likely that the duration of metaphase I varies with different environmental conditions such as temperature, food sources, and age of parents.

Oocytes were imaged at room temperature (~21°C) under a microscope (Axiovert; Carl Zeiss) attached to a spinning disk confocal head (CSU-X1; Yokogawa) controlled by Volocity (PerkinElmer). A Plan-Apochromat objective lens (63×/1.4 numerical aperture) was used with Immersol 518F oil (Zeiss). Z sections were captured at a 0.8-µm interval and are displayed after a maximum-intensity projection onto the XY plane. The regulator of chromatin condensation 1 tagged with mCherry (RCC1-mCherry) under the control of the UASp (Colombié et al., 2013) or the maternal α-tubulin at 67C promoter (this study) was used to visualize the chromosomes. α-Tubulin and microtubule-associated proteins were tagged with eGFP for visualization in live oocytes.

In this study, the morphology of the dorsal appendages was used to distinguish between stage 13 and stage 14 oocytes (King, 1970). We observed stage 13 oocytes, which have poorly developed but visible dorsal appendages with several nurse cells at their anterior end, and stage 14 oocytes with fully elongated dorsal appendages and no or few nurse cells. In Fig. S2, to observe oocytes before the NEBD, we selected late stage 12/early stage 13, which have a similar size to stage 13 oocytes but no visible dorsal appendages. These oocytes nearly always have an intact nuclear envelope and undergo NEBD within 30–60 min in most cases.

In Fig. 1 (C and D); Figs. 2, 3, and 4 B; Fig. S1 (B and C); and Fig. S3 (A and C), we show representative snapshot images of the spindle from different stage 13 and 14 oocytes. In Fig. S2 A, images were captured at specified intervals in the same late stage 12/early stage 13 oocyte, and the localization of the respective spindle proteins was followed for a maximum of 7 h 30 min. When the signal intensity of α-tubulin or spindle-associated proteins is compared, the images were taken using identical settings (the same laser power, exposure time, and camera settings). In figures where a comparison is made between stage 13 and 14 oocytes, or between a control and Mklp1/Pav or kinesin-5/Klp61F RNAi, the images of α-tubulin and spindle proteins were taken under the same conditions and shown after applying equal contrast and brightness enhancement.

To analyze spindle morphology in Figs. 1 D and 5 B, the spindle length and width were measured from the maximum-intensity projection of a Z-stack by drawing a horizontal line from one pole to the other or a vertical line at the spindle equator, respectively. A small minority of spindles (5–10%) not oriented on the XY plane were excluded from this analysis. The graphs in Figs. 1 D and 5 B show the median of the spindle length/width and the first and third quartiles of the data. Statistical significance was assessed using the two-tailed Wilcoxon rank-sum test. In Fig. 1 D, 37 stage 13 and 52 stage 14 oocytes (eight flies) from three independent experiments were pooled together and quantified. In Fig. 5 B, 32 control and 30 MKlp1/Pav RNAi oocytes from 25–30 flies were analyzed from a single experiment performed in parallel under the same conditions. This experiment has been duplicated and triplicated for control and Pav RNAi oocytes, respectively, and gave comparable results.

In Fig. 1 C, GFP-Rod or GFP-Clasp was used together with Rcc1-mCherry to estimate the frequency of oocytes in metaphase I. We define metaphase I oocytes when the chromosomes are symmetrically aligned and the kinetochore microtubules decorated by Rod/Clasp were symmetrically extended to the poles. We excluded from this analysis a small proportion of oocytes (10–20%) that showed weak signal. In Fig. S2, to estimate the timing of metaphase I, GFP-Rod or GFP-Clasp was visualized together with Rcc1-mCherry from NEBD until stable bipolar attachment was achieved. Expression of GFP-Clasp in oocytes resulted in abnormal chromosome alignment in a significant proportion of live oocytes (5 of 8), which were excluded from the analysis. These abnormalities included splitting of the chromosome mass far apart and severe misalignment of chromosomes that persisted and was not resolved over time. Because these defects were not observed in oocytes expressing any other GFP-tagged proteins, we concluded they were caused by expressing GFP-Clasp.

For measuring the intensity of α-tubulin or spindle-associated proteins (Figs. 1 D and 2; Fig. 3, A and B; Fig. 4, C and D; Fig. S2; and Fig. S3, A and C) in stage 13 and 14 oocytes, a maximum-intensity projection of each spindle was analyzed in ImageJ (National Institutes of Health). The images of each spindle were rotated so that the spindles were horizontally oriented and then cropped to a rectangular box of 30 × 10 µm (length × height) with its center positioned at the middle of the chromosome mass. In Fig. 4 C, a box of 20 × 13 µm or 32 × 20 µm was used in embryos undergoing metaphase or telophase, respectively. After separating the channels, the channel of interest was resliced from xy to xz without interpolation, and a maximum-intensity z projection was performed, after which an intensity profile was generated. For each spindle, the average background intensity was obtained by averaging the signal intensities of the first and last 2.5-µm regions. To calculate the specific signal intensity, the average background intensity was subtracted from the signal intensity at each point along the spindle. Graphs show the average specific intensity along the long axis of the spindles from images taken under the same conditions. In Fig. 3 A; Fig. 4, C and D; Fig. S2; and Fig. S3, A and C, bright foci of MKlp1/Pav outside of the spindle were manually removed using small black circles in ImageJ. In Fig. 1 D, the position of the spindle poles for each oocyte was recorded, and the length of the spindle was normalized for comparison before averaging the signal intensities. In Fig. 4 C, when the signal intensity of GFP-MKlp1/Pav was saturated, a second image was taken with a lower laser intensity. To compensate for the lower laser power in the quantifications, a laser coefficient (3.4) was estimated from the average intensity of GFP-MKlp1/Pav on three spindles without saturated signals which were taken using the two different laser powers.

The graphs in Fig. 1 F represent the averages of 17 stage 13 or 14 oocytes. Similarly, the graphs in Fig. 2; Fig. 3, A and B; and Fig. S3, A and C, represent the averages of 8–22 stage 13 or 14 oocytes. The graphs in Fig. 4 C represent the average of 30–35 telophase spindles and 10 metaphase spindles from several embryos, while the graph in Fig. 4 D represents the average of 20–21 oocytes for each condition. In Fig. S3, A and C, 11–13 spindles were analyzed for stage 13 and 14 oocytes. The error bars in the graphs (Figs. 1 F and 2; Fig. 3, A and B; Fig. 4, C and D; and Fig. S3, A and C) represent the SEM of the specific intensity at each point along the spindle axis. An unpaired two-tailed *t* test was used for testing statistical significance of differences in the signal intensity of α-tubulin. The data distribution was assumed to be normal, but this was not formally tested.

In Fig. 3 C, the frequency of spindles with Msps-GFP and TACC-GFP foci was counted in 32–44 stage 13 or 14 oocytes from five to seven flies in the same experimental conditions. Flies with fewer than four oocytes in each stage were not included in the quantification. Error bars represent the SEM of the frequency of Msps/TACC foci for each stage in the different flies, and an unpaired two-tailed *t* test was used to test the statistical significance. The data distribution was assumed to be normal, but this was not formally tested.

In Fig. 5 (C and D), we classified the spindles that had mostly bipolar morphology but with extended or partially unfocused poles as "slightly abnormal." In contrast, "very abnormal" spindles included spindles that were monopolar, multipolar, disorganized, or disintegrated into multiple spindles. Because the number of oocytes obtained from each fly was small, results from seven (Klp61F RNAi) and eight (control) different flies were combined into three batches each. The spindle abnormalities were compared between stage 13 and 14 or between control and Klp61F RNAi oocytes using an unpaired two-tailed *t* test. The data distribution was assumed to be normal, but this was not formally tested.

### Immunostaining of *Drosophila* oocytes and embryos

For immunostaining of late (stage 14) *Drosophila* oocytes, young females were matured for 3 d at 25°C with males and food supplemented with dried yeast. This allows natural ovulation and fertilization. Ovaries from 24 to 30 previously decapitated females were fixed in 100% methanol (Merck) and immunostained as follows (Cullen and Ohkura, 2001). The ovaries were sonicated with a microprobe (at 38% amplitude) using Vibra Cell (VCX500; Sonics) for 1-s bursts to remove the chorion and the vitelline membranes of the oocytes. Sonication was repeated until enough dechorionated oocytes were collected into a new tube containing fresh 100% methanol. After collection, oocytes without the chorion were rehydrated first with 500 µl of 50% methanol/50% PBS for 10 min and later in 500 µl of 100% PBS for 10 min. The oocytes were incubated for 30 min in blocking solution containing PBS-T 0.1% (PBS + 0.1% Triton X-100) and 10% FCS. After three 10-min washes with PBS-T 0.1%, the oocytes were incubated for 4 h to overnight with 100–200 µl of primary antibodies diluted in blocking solution. This was followed by three 10-min washes in PBS-T 0.1% and 2-h incubation with 100–200 µl of secondary antibody and DAPI (0.4 µg/ml) diluted in PBS-T 0.1%. Oocytes were then washed four times in PBS-T 0.1% for 1 h and

once with PBS. They were mounted in medium (85% glycerol and 2.5% propyl gallate) between a coverslip and a glass slide before sealing with nail varnish. Under this condition, a population of mostly mature stage 14 oocytes, which are naturally arrested in metaphase I (King, 1970; Theurkauf and Hawley, 1992; Page and Orr-Weaver, 1997), was obtained.

For immunostaining of syncytial embryos, females were matured for 3 d with males and yeast at 25°C, after which they were transferred to cages for embryo collection. Embryos were collected on agar plates supplemented with yeast for 1 h and aged for 1 h and 30 min. They were washed with deionized water, dechorionated in ∼10% sodium hypochlorite, and thoroughly rinsed with water. Vitelline membranes were removed by shaking for 30–60 s in methanol and heptane (1:1.3 ml). Devitellined embryos were transferred to a microfuge tube containing fresh methanol. Embryos were rehydrated through incubation with increasing concentrations of PBS in methanol (20, 40, 60, 80, and 100%) for 10 min each step, followed by immunostaining as described above for oocytes.

In Fig. 5 (A and B), even partial depletion of MKlp1/Pav caused polarity defects of oocytes, which prevented the formation of dorsal appendages that we routinely use as the marker for staging. Therefore, we had to immunostain a mixed population of oocytes predominantly, but not exclusively, in stage 14. This is likely to be the reason we observed larger variations in spindle lengths.

The following primary antibodies were used for immunostaining of *Drosophila* oocytes and embryos: anti–α-tubulin (mouse monoclonal DM1A; 1:250; Sigma-Aldrich), anti-Pav (rabbit polyclonal; 1:40; Ohkura Lab; this study), and anti-GFP (rabbit polyclonal; 1:250; A11122; Thermo Fisher Scientific). Alexa Fluor 488–, Cy3-, and Cy5-conjugated secondary antibodies were used (1:250 to 1:1,000; Jackson Laboratory or Molecular Probes), and DNA was stained using 0.4 µg/ml DAPI (Sigma-Aldrich).

After immunostaining, *Drosophila* oocytes were imaged as previously described using an Axiovert 200M microscope (Zeiss) attached to a confocal laser scanning head LSM 800 (Zeiss; Beaven et al., 2017; Romé and Ohkura, 2018). Oocytes were visualized using a Plan-Apochromat objective lens (63×/1.4 numerical aperture) with Immersol 518F oil (Zeiss). Z-sections were captured with 0.5-µm interval, ∼1-µm optical section, 512 × 512-pixel/zoom 2 (∼0.1 µm/pixel), average of 4. The maximum-intensity projection of multiple Z-planes covering the entire spindle is shown in all figures.

### Immunoblotting of *Drosophila* oocytes

For immunoblotting of oocytes, the ovaries of females matured at 25°C for 3 d with males and dried yeast were dissected in methanol, and 100 oocytes were collected into a microfuge tube. To prepare protein samples for immunoblots, methanol was removed and replaced with boiling sample buffer (50 mM Tris-Cl, pH 6.8, 2% SDS, 10% glycerol, 0.1% bromophenol blue, and 715 mM 2-mercaptoethanol). After the mixture was boiled for 5 min to denature proteins, oocytes were crushed in the microtube using a pestle (Eppendorf). The equivalent of 15 oocytes (Fig. S3, B and D–F) was loaded on a gel for SDS-PAGE. The

proteins were then transferred onto nitrocellulose membranes (Protran 0.2 NC; GE Healthcare) and stained for total proteins using a Reversible Protein Stain kit for Nitrocellulose Membranes (Thermo Fisher Scientific). The membrane was incubated with primary antibodies followed by fluorescent secondary antibodies (LI-COR) and visualized on an Odyssey CLx imaging scanner (v3.0.30; LI-COR). The brightness and contrast settings were adjusted equally in the entire field without removing or modifying features.

Antibodies used for immunoblotting were anti-Pav (rabbit polyclonal; 1:250; this study) and anti–α-tubulin (mouse monoclonal DM1A; Sigma-Aldrich; 1:2,000). The secondary antibodies used were IRDye 800CW–conjugated goat anti-rabbit (1:20,000) and IRDye 680LT–conjugated goat anti-mouse (1:15,000) from LI-COR. The anti-Pav antibody was generated in rabbit using bacterially produced GST-Pav as the antigen, as previously described (Romé and Ohkura, 2018).

### Online supplemental material
Fig. S1 shows the localization of spindle proteins in stage 13 and 14 oocytes. Fig. S2 shows time-lapse imaging of spindle proteins. Fig. S3 shows GFP-MKlp1/Pav expressed from various constructs.

## Acknowledgments
We are grateful to the members of the Ohkura laboratory for their help and R. Beaven, S. Beard, A Gluszek, H. Child, and F. Cullen for their contributions and for generating reagents; and D. Glover and R. Karess for fly stocks and reagents.

The Bloomington Drosophila Stock Center/Resource Center (National Institutes of Health grants P40OD018537 and 2P40OD010949-10A1) and the Transgenic RNAi Project at Harvard Medical School (National Institutes of Health/National Institute of General Medical Sciences grant R01-GM084947) provided fly stocks and reagents. This work is supported by the Wellcome Trust (081849, 098030, 206315, 099827, 092076, and 203149) and Biotechnology and Biological Sciences Research Council (BB/S013059).

The authors declare no competing financial interests.

Author contributions: M.F.A. Costa and H. Ohkura designed and performed experiments, analyzed the data, and wrote the manuscript.

Submitted: 18 February 2019

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
