## [Reviewer comments · The Journal of Cell Biology]

The molecular architecture of the meiotic spindle is remodelled during metaphase arrest in oocytes

Mariana Costa and Hiroyuki Ohkura

Corresponding Author(s): Hiroyuki Ohkura, The University of Edinburgh

Review Timeline:

Submission Date:	2019-02-18
Editorial Decision:	2019-03-18
Revision Received:	2019-05-21
Editorial Decision:	2019-06-13
Revision Received:	2019-06-17

Monitoring Editor: Arshad Desai

Scientific Editor: Andrea Marat

Transaction Report:

DOI: <https://doi.org/10.1083/jcb.201902110>

March 18, 2019

Re: JCB manuscript #201902110

Prof. Hiroyuki Ohkura
The University of Edinburgh
Wellcome Centre for Cell Biology
School of Biological Sciences
Max Born Crescent
Edinburgh EH9 3BF
United Kingdom

Dear Dear Hiro:

Thank you for submitting your manuscript entitled "The molecular architecture of the meiotic spindle is remodelled during metaphase arrest in oocytes" to Journal of Cell Biology. The manuscript has now been assessed by expert reviewers, whose reports are appended below. Unfortunately, after an assessment of the reviewer feedback, our editorial decision is against publication in JCB.

Regretfully, after considering the reviewer feedback, we are unable to consider the manuscript further for JCB. While the characterization of localization of a number of proteins on oocyte spindles at different stages was uniformly appreciated, the reviewers are not persuaded that the analysis is focused on an event associated with spindle stabilization during oocyte metaphase arrest and they also have concerns related to the Pav4/Mklp1 inhibition data and its interpretation. Given these dual concerns, the reviewers did not indicate the level of enthusiasm that we would need for the work to be considered further. We hope that the reviewer feedback will prove helpful to you and your colleagues in moving forward with submitting this manuscript to another journal.

I do realize that significant further work and expansion might convincingly address some of these issues, but I am hesitant to encourage you to work towards the aim of further consideration at JCB. The level of reviewer criticism makes it impossible for me to guarantee that we will be able to invite resubmission, even after revision. Therefore, it does seem that it will be best for you to consider another journal for this work. Our journal office will transfer your reviewer comments to another journal upon request.

I am sorry our decision is not more positive, but hope that you find the reviews constructive. Of course, this decision does not imply any lack of interest in your work and we look forward to future submissions from your lab.

Thank you for your interest in Journal of Cell Biology.

Sincerely,

Arshad Desai, PhD
Monitoring Editor

Andrea L. Marat, PhD

Reviewer #1 (Comments to the Authors (Required)):

The molecular architecture of the meiotic spindle is remodelled during metaphase arrest in oocytes

Costa et al. study the dynamic behavior of the spindle and of multiple spindle-associated proteins through live cell imaging of GFP-tagged proteins during two stages of oocyte meiosis in *Drosophila*. Their main conclusion is that some studied proteins, but not all of them, display a dynamic relocalization within the spindle between stage 13 and stage 14 oocytes. From this analysis, the authors conclude that their study provides a novel insight into the mechanism that ensures spindle stability and bipolarity during the long metaphase arrest that oocytes of most species undergo before fertilization.

Overall, this study represents an impressive amount of live imaging experiments on a very difficult model system, the *Drosophila* oocyte. The study is however mostly descriptive and lacks functional details that would explain the dynamic relocalization of some of the tagged-proteins analyzed here.

Major comments:

-My main concern is about the author's central claim that comparing stage 13 and stage 14 oocytes allows drawing conclusions on the behavior of the spindle during the long metaphase I arrest. Since the status of the kinetochore-microtubule attachments changes between stage 13 and stage 14 (page 6, line 3), I would argue that the authors are in fact comparing prometaphase I to metaphase I oocytes. I therefore disagree with their use of the terms 'early' and 'late' metaphase I. This would make this study more relevant to the mechanisms of oocyte spindle assembly rather than to spindle structure maintenance during the metaphase I arrest. Unless the authors have a good reason to think this is not the case, I would suggest refocusing the manuscript on spindle assembly mechanisms rather than on spindle bipolarity and stability maintenance during the metaphase I arrest.

Minor comments:

-Could the authors explain why the timings of the analyzed spindles on figure S3 are so different (between 2h and 7:30h)?

-The scaling of some images is a bit strange. For example, how can the background of the 2 first GFP-Mklp1 time points (Figure S3) be so dark and become so gray after 2h? This suggests that the scaling was adjusted (with auto-scaling or manually) over time in these images.

-Figure S3 top left panel legend: 'cchromosomes'

Reviewer #2 (Comments to the Authors (Required)):

In this study, the authors characterize the localization of 24 GFP-tagged spindle proteins on early vs late metaphase-arrested *Drosophila* oocyte meiotic spindles by live imaging and demonstrate that 13 proteins change their localization in a manner that correlates with a shortening and

fattening of spindles between these 2 stages. The authors further show that phosphorylation of the kinesin-6, PavKLP, is required to keep PavKLP off of the early spindles and is required for the transition to short fat spindle shape in late metaphase arrested oocytes. The data is of high quality and it is very impressive that 24 proteins were localized in living oocyte meiotic spindles. The significance proposed by the authors is that the changing localization of these proteins is required to stabilize meiotic spindles during the prolonged metaphase arrest that is common in the oocytes of many species. There are 2 missing links to strong significance that would be required for publication in JCB. First, the late PavKLP-depleted spindles that have "early" morphology should be falling apart, only after prolonged arrest, if the transition to "late" morphology is required for spindle stability during prolonged arrest. No such data is presented. Because PavKLP is required for spindle assembly and likely for anaphase/telophase, the authors would need to devise a method to deplete or inhibit PavKLP only during metaphase, then assay the successful completion of meiosis in order to demonstrate the real significance of this "metaphase stabilization pathway". Second, the work is only significant if the metaphase arrest under study is physiological rather than an artifact of removing oocytes from their mother (See below).

Details

For the reader to understand the significance of this work, more *Drosophila* specific information is needed in the introduction. The introduction gives a reference for the fact that human metaphase II arrest can last 24 hrs. However, there is no way for the reader to guess the time between *Drosophila* stage 13 early metaphase arrest and stage 14 late metaphase arrest. The single time-lapse sequence of GFP:tubulin in Fig. S3 suggest the change in shape of the spindle takes 4 hrs but there is no mention of this conclusion or how it would fit with the normal timing of the reproductive cycle. In the referenced Gilliland 2009 paper, virgin females were used to enrich for late metaphase embryos. This would suggest that stage 14 embryos from virgin females have likely been arrested for a longer time than would ever occur naturally. The methods for this paper do not state whether males were present. If the described morphological changes occur during a natural reproductive cycle and are necessary for successful reproduction, then there would be significance to this work. If the described morphological changes only occur during an unnatural extension of the duration of metaphase arrest, there is very little significance to this work. If the manuscript does not show that the described morphological changes are required for successful reproduction, the significance would also be diminished.

The authors should state whether PavKLP-4A causes early spindles to have a short fat shape. Either way, this would suggest whether or not PavKLP dephosphorylation is sufficient to drive the spindle shape change.

Figure 5 shows the effect of Pav KLP depletion on stage 14 spindle length and width. Since the authors conclude that Pav KLP is required for the change from stage 13 morphology to stage 14 morphology, it would be important to show data for stage 13 control and Pav KLP RNAi. If stage 13 PavKLP-depleted spindles are longer and skinnier than control, the conclusion about PavKLPs role in shape change would have to be modified. Since the authors argue that recruitment of Pav KLP to meiotic spindles is important for stabilizing spindles during prolonged metaphase arrest, the reader would like to see stage 14 PavKLP RNAi spindles look normal at early stages, then fall apart after prolonged arrest.

Gilliland 2009 suggests that you might find chromosome 4 in the out position in stage 13 more often than in the stage 14. Since chromosomes might impact the shape of the spindle and vice versa, some mention of 4th chromosome position should be made here.

Legends for Fig. 2, Fig. S1 and Fig. 4 need to state the number of oocytes analyzed and included in the intensity plots.

The section on the differences in timing of accumulation of different proteins determined by time lapse imaging is weak without stating an n for each time-lapse result.

Reviewer #3 (Comments to the Authors (Required)):

This paper by Costa and Ohkura aims to address the question of how spindles are remodeled during metaphase arrest in *Drosophila* oocytes. The authors generate an array of transgenic flies, and compare spindles in stage 13 oocytes (i.e. early metaphase arrest) to those in stage 14 oocytes (i.e. late metaphase arrest). These studies reveal that a number of spindle proteins relocalize during this period - some spindle proteins become enriched at the midzone (e.g. microtubule crosslinkers such as PRC1 and MKlp1) while others concentrated more strongly at spindle poles in later oocytes. The authors also investigate the mechanism by which MKlp1 accumulates at the spindle equator, and perform depletion experiments that suggest that MKlp1 may contribute to the spindle shortening that occurs between stage 13 and 14.

As little is known about how extended metaphase arrest affects oocyte spindles in any organism, this paper addresses an important biological problem. Moreover, the experiments are well-executed, and represent an impressive amount of effort, with a large number of spindle proteins examined. While this paper is mostly descriptive and only has a few functional experiments, the JCB report format allows manuscripts where the underlying mechanism is less developed - therefore, in principle this paper could be appropriate for publication in this format. However, in my opinion a number of concerns (major points, detailed below) preclude publication of this paper in its current form.

Major points:

1. Question about oocyte staging and the classification of "metaphase arrest": This manuscript claims to address the question of how the spindle is remodeled during metaphase arrest. It is true that little is known about this in any system, and therefore this topic is of interest. However, I had some questions about the staging of the oocytes that led me to wonder if the conclusions were properly interpreted. I will admit that I do not work in this system (and so the authors may be able to provide me with information that would mitigate my concern), but my understanding is that stage 13 oocytes represent a mixture of prometaphase and metaphase spindles. Thus, it has been proposed that prometaphase and metaphase spindles can be distinguished by chromosome organization, with chromosomes transitioning from more extended in prometaphase to more round/compact in metaphase (e.g. Gilliland, *Dev Biol* 2009; Radford and McKim, *Jove* 2016). Since some of the spindles shown in the figures appear to have a more extended karyosome or chromosomes off the metaphase plate (e.g. the image in Figure 1B, some of the images in Figure 4B and S1, the image in S2C, and many of the images in S3), it made me wonder if all of the stage 13 oocytes that were analyzed were really in metaphase. If not, then the spindle reorganization that the authors describe might represent changes associated with the transition from prometaphase to metaphase, rather than changes associated with an extended metaphase arrest. Although it is still important to understand how spindles transition from prometaphase to metaphase in this system, it would affect the interpretation of the experiments and would make the findings less relevant to understanding metaphase arrest in other systems. Therefore, I would ask that the

authors better explain and justify their staging of the oocytes. If all of the stage 13 oocytes are not in metaphase, then I think that some of the conclusions need to be revised, and this would decrease the impact of the manuscript.

2. Overinterpretation of MKlp1/Pav knockdown experiments (Figure 5A, B): The partial Mklp1 depletion experiments are not particularly convincing. Although I realize that a statistical test supports the author's conclusion that the MKlp1 spindles are longer, there is so much variability in spindle length in both the control and the MKlp1 depletion conditions that it was difficult to interpret these experiments. In both conditions, you get spindles ranging from as short as ~8um to spindles longer than 30um. Since this is such a huge range, can you really conclude that MKlp1 depletion makes the spindles look more like stage 13 spindles? I think part of the problem is the control for this experiment. I would expect the range of spindle lengths in the control to look like the stage 14 spindle lengths in Figure 1C (ranging from ~5um to 15um) - why are the control lengths in Figure 5B so much different from the stage 14 lengths in Figure 1C? Given these issues, I thought that the author's conclusions based on this experiment (page 9, lines 1-2: "accumulation of MKlp1/Pav to the spindle equator plays an important role in shortening and widening of late metaphase spindles in stage 14" and a similar statement on page 9, lines 19-21) were not well justified. And since this was one of the few functional experiments in the paper, if the authors can't provide additional support for this conclusion, then I think that would decrease the impact of the paper.

Minor points:

- Figure S1B: I realize this is not a focus of the paper and the authors are not intending on pursuing the proteins shown further, but I found some of the images of kinetochore proteins in Figure S1B hard to interpret. In particular, the Clasp image is confusing, as there aren't just kinetochore dots in stage 13 oocytes, but streaks (and these streaks are concentrated on one side - why do they look like that?). Also the CycA dots don't all seem to be chromosome associated. What are these extra dots? I would suggest adding a few sentences to the figure legend to explain the images so readers not used to looking at Drosophila oocytes could better interpret them.

- Page 6, line 9: The sentence references Figure 1C, but I don't think that is the right figure callout.

Typos:

- Page 3, line 22: "localization of the spindle proteins" should be "localization of spindle proteins"

- Page 4, line 24: "stabilizing the spindle bipolarity" should be "stabilizing spindle bipolarity" or "stabilizing the bipolar spindle"

- Page 9, line 9: "lacks" should be "lack"

21st May 2019

Dear Editor,

We **re-submit** our manuscript (the original manuscript #201902110) entitled "The molecular architecture of the meiotic spindle is remodelled during metaphase arrest in oocytes" for publication as a *Report* in *J. Cell Biol.* Although the three reviewers for the original manuscript have appreciated its high technical quality and potential significance, all expressed two same major concerns. As these concerns were fundamental, **we carried out additional experiments, which have led to significant findings that fully address these two major concerns.** We have incorporated these findings and revised the manuscript extensively. We very thankful for the reviewers' comments, which are very constructive and helpful. We would be grateful if you could reconsider the manuscript for publication in *J. Cell Biol.*

(1) prometaphase I vs metaphase I: The reviewers' first major concern was that the observed changes of localisation may take place during prometaphase I, rather than metaphase I arrest. By further time-lapse live imaging, we determined the timings of localisation changes relative to metaphase I onset. **MKlp1/Pav, Kinesin-5/Klp61F and Augmin change localisation during the metaphase I arrest.** In contrast, Sentin and TPX2/Mei-38 accumulated to the spindle poles during prometaphase I. Therefore, we made significant changes in the text to distinguish between changes in prometaphase I and metaphase I. Moreover, **we have estimated the normal length of metaphase I arrest in flies under our experimental conditions.** In this report, we focus our further mechanistic and functional studies on two microtubule cross-linking kinesins, MKlp1/Pav and Kinesin-5/Klp61F, which accumulate to the spindle equator during metaphase I arrest.

(2) functional significance: Their second major concern was that we have not demonstrated the functional significance of these localisation changes. Strikingly, we have now found that **knockdown of Kinesin-5/Klp61F did not disrupt bipolar spindles during prometaphase I, but led to a failure to maintain bipolar spindles only during metaphase I arrest.** This suggests that the change of Kinesin-5/Klp61F localisation to the equator during the arrest is critical for the stability of arrested spindles. Therefore, this new finding provides a crucial functional evidence that supports the importance of the dynamic relocalisation of spindle proteins during the arrest.

Our new findings clarify the timings of localisation changes and provide an evidence for functional significance in maintaining metaphase spindles. In addition, we have addressed other issues the reviewers pointed out for the original version (see point-to-point responses in the following pages). Incorporating reviewers' comments has greatly strengthened our manuscript. Therefore, we are confident that this new version is suitable for publication in *J Cell Biol.*

Best regards,

Hiro Ohkura
Professor of Cell Biology and Wellcome Investigator in Science
Wellcome Centre for Cell Biology
The University of Edinburgh
Edinburgh EH9 3BF, UK
+44-131-650-7094, h.ohkura@ed.ac.uk

[Major additions or changes]

- Including results from Kinesin-5/Klp61F depletion, showing spindle defects specific to metaphase I arrest (Fig.5C,D).
- Including further time-lapse imaging data to determine the timing of relocalisation relative to metaphase I onset (Fig.S3).
- Including information on long prometaphase I in oocytes in the introduction, and clarify wordings on the timing of relocalisation in the text.
- Including more information on the condition of maturation, and a rough estimate of the length of metaphase I in flies.
- Including sample numbers.
- Revising the text for MKlp1/Pav RNAi to reflect some uncertainty.
- Including comments on the localisation of some kinetochore proteins in the legend of Fig S1.
- Correcting typos.

[point-to-point response to comments from Reviewer 1]

The reviewer considers that *"this study represents an impressive amount of live imaging experiments on a very difficult model system"*. However, s/he has two major reservations, which we have addressed by additional experiments.

"The study is however mostly descriptive and lacks functional details that would explain the dynamic relocalization of some of the tagged-proteins analyzed here."

To gain an insight into functional significance of the relocalisation, we carried out additional experiments. Firstly, we tried, without success, different conditions and RNAi lines of MKlp1/Pav to find a degree of depletion which is enough to go through early oogenesis, but disrupts the spindle integrity.

Strikingly, when we depleted Kinesin-5/Klp61F, which accumulates to the equator in late metaphase I, we found that spindles are initially bipolar in prometaphase I but fail to maintain bipolar spindles only during metaphase I arrest. This suggests that late localisation of Kinesin-5/Klp61F to the equator is important for spindle integrity during the metaphase arrest. Therefore, this new finding provides crucial functional evidence that supports the importance of the dynamic relocalisation of spindle proteins during the arrest. We have included this new finding with figures (Fig.5C,D).

"My main concern is about the author's central claim that comparing stage 13 and stage 14 oocytes allows drawing conclusions on the behavior of the spindle during the long metaphase I arrest."

This concern is well justified. We agree that comparisons between stage 13 and 14 do not distinguish changes between prometaphase I and metaphase I, or between early and late metaphase I. We have carried out further live imaging to determine the timings of metaphase onset and relocalisation of representative spindle proteins. It showed that MKlp1/Pav, Kinesin-5/Klp61F and Augmin change their localisation during metaphase I arrest, while Sentin and TPX2/Mei-38 change during prometaphase I. We have included this new finding with figures (Fig.S3) and clarified the text.

Minor comments:

"Could the authors explain why the timings of the analyzed spindles on figure S3 are so different (between 2h and 7:30h)?"

This is because we usually terminated imaging when it reached the final steady state.

"The scaling of some images is a bit strange. For example, how can the background of the 2 first GFP-Mklp1 time points (Figure S3) be so dark and become so gray after 2h? This suggests that the scaling was

adjusted (with auto-scaling or manually) over time in these images."

Within each time series, identical imaging conditions were used, and the contrast has been adjusted equally for presentation. We do not know the reason for sure, but the spindle may have changed the position in the Z axis, which altered the apparent cytoplasmic signal intensity.

"Figure S3 top left panel legend: 'cchromosomes' "

Thank you very much for pointing this out. We have corrected it.

[Reviewer 2]

This reviewer considers that *"The data is of high quality and it is very impressive that 24 proteins were localized in living oocyte meiotic spindles."* However, s/he also pointed out *"there are 2 missing links to strong significance that would be required for publication in JCB."* We have filled these two gaps by further experiments

"First, the late PavKLP-depleted spindles that have "early" morphology should be falling apart, only after prolonged arrest, if the transition to "late" morphology is required for spindle stability during prolonged arrest. No such data is presented."

We can only partially deplete MKlp1/Pav because full depletion results in severe oogenesis defects. We tried, without success, different conditions and RNAi lines of MKlp1/Pav to find a degree of depletion which is enough to go through early oogenesis, but disrupts the spindle integrity.

Strikingly, when we depleted Kinesin-5/Klp61F, which accumulates to the equator in late metaphase I, we found that spindles are initially bipolar in prometaphase I but fail to maintain bipolar spindles only during metaphase I arrest. This suggests that the transition of Kinesin-5/Klp61F to late localisation to the equator is important for spindle stability during the metaphase arrest. Therefore, this new finding provides crucial functional evidence that supports the importance of the dynamic relocalisation of spindle proteins during the arrest. We have included this new finding with figures (Fig.5C,D) and clarified the text.

"Second, the work is only significant if the metaphase arrest under study is physiological"

In this study, females were matured in the presence of males and food (with yeast), which stimulates oogenesis and ovulation. Based on our observations, we roughly estimate that metaphase I lasts ≥ 6 hours at 21°C under our experimental conditions (** see below). In a natural habitat where food or males are not always available, oocytes in flies often arrest in metaphase I even longer (up to a few days). Therefore, fly oocytes must have adapted to a long metaphase I arrest, which they can encounter in different natural habitats.

In our study, we imaged oocytes immediately after dissection from their mothers to determine protein localisation and define prometaphase I or metaphase I. Importantly, our main conclusions on the localisation changes were deduced from this analysis. The changes of protein localisation were later confirmed by time-lapse live imaging of oocytes, which typically lasted for 6-7 hours after dissection from their mothers. Therefore, our results must reflect the real dynamics of protein localisation during natural metaphase arrest, not artefacts resulting from long incubation outside of the mothers.

Furthermore, images of oocytes taken immediately after dissection showed different requirement of Kinesin-5/Klp61F between stage 13 (prometaphase I and early metaphase I) and stage 14 (late metaphase I). Combined with the late accumulation of Kinesin-5 to the equator, this suggests the presence of a specific mechanism to stabilise the spindle during the natural metaphase arrest.

We have included more information in Results and Methods to clarify these points. We have also included a diagram (Fig.1B) that summarises the estimated timing of various events during stage 13 and 14.

(**** Estimation of the metaphase I length.** Mature female flies carry more stage-14 oocytes than stage-13 oocytes. Immediately after dissection, 26-40% of spindles in stage-13 oocytes are in prometaphase I. In our time-lapse imaging at 21°C, prometaphase I lasted about 2 hours, with bipolar spindles present for 1.5 hours. From these, we roughly estimated that metaphase I lasts ≥ 6 hours at 21°C under our experimental condition. It is likely that the duration of metaphase I varies with different environmental conditions such as temperature, food sources, and age of parents.)

"The authors should state whether PavKLP-4A causes early spindles to have a short fat shape. Either way, this would suggest whether or not PavKLP dephosphorylation is sufficient to drive the spindle shape change."

As GFP-MKlp1/Pav does not localise or only weakly localises to early spindles and -4A localises only near the equator, it is difficult to define and compare the spindle morphologies between the two in live oocytes where we can distinguish between early and late spindles.

"it would be important to show data for stage 13 control and Pav KLP RNAi. If stage 13 PavKLP-depleted spindles are longer and skinnier than control. ...the reader would like to see stage 14 PavKLP RNAi spindles look normal at early stages, then fall apart after prolonged arrest."

Unfortunately, even partial depletion of MKlp1/Pav causes oocyte polarity defects, including a failure to form dorsal appendages which we use as the marker of staging. Therefore, we observed a mixed population which is predominantly in stage-14. We have clarified this point, and revised the text to reflect the uncertainty. We tried, without a success, different conditions and RNAi lines of MKlp1/Pav to find a degree of depletion which is enough to go through early oogenesis, but disrupts the spindle integrity.

Instead, we found that depletion of Kinesin-5/Klp61F, which accumulates to the equator in late metaphase I, results in bipolar spindles which are bipolar during prometaphase I but fall apart/collapse during metaphase I arrest. We have included this new finding with figures and clarified the text.

"Gilliland 2009 suggests that you might find chromosome 4 in the out position in stage 13 more often than in the stage 14. Since chromosomes might impact the shape of the spindle and vice versa, some mention of 4th chromosome position should be made here."

In time-lapse live imaging, we often see chromosome 4 still separated from the main chromosome mass even 6 hours after NEBD (~4 hours after metaphase onset). We suspect that congression of chromosome 4 may need a longer arrest. However, as it is challenging to identify chromosome 4 reliably and confidently in this live imaging, we would like to avoid inaccurate or misleading statements on chromosome 4.

"Legends for Fig. 2, Fig. S1 and Fig. 4 need to state the number of oocytes analyzed and included in the intensity plots."

The sample numbers are now included in the figure legends.

"The section on the differences in timing of accumulation of different proteins determined by time lapse imaging is weak without stating an n for each time-lapse result."

Complete time-lapse imaging of at least three oocytes have been carried out for each of GFP-tagged proteins. As these are technically challenging experiments, sample numbers are still small, but the results agree with each other and are consistent with snapshot imaging. In Fig. S3B, we have included the actual times of the changes from all oocytes analysed for each spindle protein.

[Reviewer 3]

This reviewer considers that *"this paper addresses an important biological problem. Moreover, the experiments are well-executed, and represent an impressive amount of effort"*. S/he thinks that *"in principle this paper could be appropriate for publication" in this format, but that "a number of concerns preclude publication of this paper in its current form"*, which we have addressed by further experiments.

Major points:

"Question about oocyte staging and the classification of "metaphase arrest""

The reviewer's concern is well justified. Indeed, comparisons between stage 13 and 14 do not distinguish changes between prometaphase I and metaphase I, or between early and late metaphase I. Through our further experiments monitoring microtubule attachment to kinetochores, we also estimated that stage-13 oocytes represent a mixture of prometaphase and metaphase spindles, and stage 14 represent metaphase spindles only (Fig 1C). We carried out further time-lapse live imaging to determine the timing of the metaphase I onset (defined by full bipolar attachment), and relocalisation of representative GFP-tagged proteins. It showed that MKlp1/Pav, Kinesin-5/Klp61F and Augmin change their localisation during metaphase I arrest, while Sentin and TPX2/Mei-38 change in prometaphase I. We have included this new finding with figures (Fig.S3) and clarified the text.

"Overinterpretation of MKlp1/Pav knockdown experiments"

MKlp1/Pav RNAi causes polarity defects of oocytes, which prevents the formation of dorsal appendages that we used as the marker for staging. Therefore, we have to immunostain a mixed population of oocytes which is predominantly, but not exclusively, in stage 14. This is likely to be the reason why we observed larger variations in spindle lengths, and makes our results less conclusive. The text has been revised to reflect this uncertainty.

"if the authors can't provide additional support for this conclusion, then I think that would decrease the impact of the paper."

We now found that depletion of Kinesin-5/Klp61F, which localises to the equator only in late metaphase I, results in normal bipolar spindles in prometaphase /early metaphase I (stage 13), but they fail to maintain bipolarity only during metaphase I arrest. This suggests that the change of localisation of Kinesin-5/Klp61F to the equator is important to maintain a bipolar spindle in the metaphase arrest. Therefore, this new finding provides crucial functional evidence to support the importance of remodelling of metaphase spindles during the arrest. We have included this new finding with figures (Fig.5C,D) and clarified the text.

Minor points:

"Figure S1B: I would suggest adding a few sentences to the figure legend to explain the images so readers not used to looking at Drosophila oocytes could better interpret them."

We have moved Clasp data to Fig.1 which is used for determining the timing of metaphase onset. In prometaphase, Clasp first accumulates to kinetochores (unattached), and then localises to kinetochore microtubules (attached) which are not symmetric, before all kinetochore microtubules are symmetrically attached (bipolar attachment; at onset of metaphase I). Cyclin A weakly localises to kinetochores. The dots apparently not associated to chromosomes in this image are likely to be the kinetochores of small chromosome 4. We include more information in the figure legends.

"Page 6, line 9: The sentence references Figure 1C, but I don't think that is the right figure callout."

We have corrected them.

"Typos:

- Page 3, line 22: "localization of the spindle proteins" should be "localization of spindle proteins"
- Page 4, line 24: "stabilizing the spindle bipolarity" should be "stabilizing spindle bipolarity" or "stabilizing the bipolar spindle"
- Page 9, line 9: "lacks" should be "lack" "

Thank you very much for pointing out these typos. We have corrected them.

June 13, 2019

RE: JCB Manuscript #201902110R-A

Prof. Hiroyuki Ohkura
The University of Edinburgh
Wellcome Centre for Cell Biology
School of Biological Sciences
Max Born Crescent
Edinburgh EH9 3BF
United Kingdom

Dear Prof. Ohkura:

Thank you for submitting your revised manuscript entitled "The molecular architecture of the meiotic spindle is remodelled during metaphase arrest in oocytes". You will see that while the reviewers find your study significantly improved, they have brought up a few remaining issues. Overall, we find that your functional analysis has been improved and that your revised study describes transactions that will be of interest. Therefore, we think these remaining issues can be addressed via a thorough and careful editing of your text and figures in response to all remaining reviewer concerns. Assuming this is done satisfactorily, we would be happy to publish your paper in JCB pending final revisions necessary to meet our formatting guidelines (see details below).

A. MANUSCRIPT ORGANIZATION AND FORMATTING:

Full guidelines are available on our Instructions for Authors page, <http://jcb.rupress.org/submission-guidelines#revised>. **Submission of a paper that does not conform to JCB guidelines will delay the acceptance of your manuscript.**

1) Text limits: Character count for Reports is < 20,000, not including spaces. Count includes title page, abstract, introduction, results and discussion, acknowledgments, and figure legends. Count does not include materials and methods, references, tables, or supplemental legends.

2) Figures limits: Reports may have up to 5 main text figures.

3) Figure formatting: Scale bars must be present on all microscopy images, including inset magnifications. Molecular weight or nucleic acid size markers must be included on all gel electrophoresis.

4) Statistical analysis: Error bars on graphic representations of numerical data must be clearly described in the figure legend. The number of independent data points (n) represented in a graph must be indicated in the legend. Statistical methods should be explained in full in the materials and methods. For figures presenting pooled data the statistical measure should be defined in the figure legends. Please also be sure to indicate the statistical tests used in each of your experiments

(either in the figure legend itself or in a separate methods section) as well as the parameters of the test (for example, if you ran a t-test, please indicate if it was one- or two-sided, etc.). Also, if you used parametric tests, please indicate if the data distribution was tested for normality (and if so, how). If not, you must state something to the effect that "Data distribution was assumed to be normal but this was not formally tested."

5) Abstract and title: The abstract should be no longer than 160 words and should communicate the significance of the paper for a general audience. The title should be less than 100 characters including spaces. Make the title concise but accessible to a general readership.

6) Materials and methods: Should be comprehensive and not simply reference a previous publication for details on how an experiment was performed. Please provide full descriptions in the text for readers who may not have access to referenced manuscripts.

7) Please be sure to provide the sequences for all of your primers/oligos and RNAi constructs in the materials and methods. You must also indicate in the methods the source, species, and catalog numbers (where appropriate) for all of your antibodies. Please also indicate the acquisition and quantification methods for immunoblotting/western blots.

8) Microscope image acquisition: The following information must be provided about the acquisition and processing of images:

- a. Make and model of microscope
- b. Type, magnification, and numerical aperture of the objective lenses
- c. Temperature
- d. Imaging medium
- e. Fluorochromes
- f. Camera make and model
- g. Acquisition software
- h. Any software used for image processing subsequent to data acquisition. Please include details and types of operations involved (e.g., type of deconvolution, 3D reconstitutions, surface or volume rendering, gamma adjustments, etc.).

10) Supplemental materials: There are strict limits on the allowable amount of supplemental data. Reports may have up to 3 supplemental display items (figures and tables). Please also note that tables, like figures, should be provided as individual, editable files. A summary of all supplemental material should appear at the end of the Materials and methods section.

13) ORCID IDs: ORCID IDs are unique identifiers allowing researchers to create a record of their various scholarly contributions in a single place. At resubmission of your final files, please consider providing an ORCID ID for as many contributing authors as possible.

B. FINAL FILES:

-- High-resolution figure and video files: See our detailed guidelines for preparing your production-ready images, <http://jcb.rupress.org/fig-vid-guidelines>.

Thank you for this interesting contribution, we look forward to publishing your paper in Journal of Cell Biology.

Sincerely,

Arshad Desai, PhD
Monitoring Editor

Andrea L. Marat, PhD
Scientific Editor

Journal of Cell Biology

Reviewer #1 (Comments to the Authors (Required)):

In a previous version of this manuscript, Costa et al compared the dynamic behavior of several spindle-associated factors in stage 13 and stage 14 *Drosophila* oocytes. They assumed that all analyzed oocytes were arrested in metaphase I of meiosis, and that any observed change in the localization of a component would reflect a dynamic remodeling of the spindle (and associated factors) during the MI arrest. However, concerns were raised by all three reviewers that oocytes in stage 13 were perhaps not in metaphase I but rather still in prometaphase I. In this revised version of their manuscript, the authors provide a significant amount of new data that help clarifying the cell cycle stage of the analyzed oocytes and supports their main conclusions. Some parts of the manuscript would still benefit from a significant rewriting/reorganization for better clarity, but overall all my concerns and comments have been addressed.

Reviewer #2 (Comments to the Authors (Required)):

In this study, the authors characterize the localization of 24 GFP-tagged spindle proteins on early vs late metaphase-arrested *Drosophila* oocyte meiotic spindles by live imaging and demonstrate that 13 proteins change their localization in a manner that correlates with a shortening and fattening of spindles between these 2 stages. The authors further show that phosphorylation of the kinesin-6, PavKLP, is required to keep PavKLP off of the early spindles and is required for the transition to short fat spindle shape in late metaphase arrested oocytes. The data is of high quality and it is very impressive that 24 proteins were localized in living oocyte meiotic spindles. The significance proposed by the authors is that the changing localization of these proteins is required to stabilize meiotic spindles during the prolonged metaphase arrest that is common in the oocytes of many species. My main concerns from the previous review remain. The introduction cites metaphase arrest durations of hours but cites no specific reference for the length of the arrest in *Drosophila*. The only units of time are in a diagram in Fig. 1B and in single time-lapse sequences shown in Fig S3. The source of the times is referenced as Materials and Methods. The text refers to Fig. S3 as a very small number of time-lapse sequences. Thus 95% of the paper relies on determining the time in metaphase from the morphology of dorsal appendages. Since the paper is about stabilization of spindles during many hours of arrest, the lack of units of hours in the data weakens the overall significance. A previous concern remains is that there is no clean experiment showing that any molecule is required for stabilization for hours without affecting initial assembly.

Reviewer #3 (Comments to the Authors (Required)):

The revised manuscript by Costa and Ohkura is greatly improved, as the authors have made a major effort to address the concerns raised by the reviewers. The addition of the kinesin-5 depletion experiment adds more mechanistic insight to the paper, in my opinion increasing impact. Further, the new experiments to clarify staging of oocytes increases confidence that the authors are documenting changes that occur during prolonged metaphase arrest, as they originally claimed. In addition, as before, the current version of this manuscript represents a large amount of work, the experiments are well-executed, and the biological problem is important.

However, there are a few remaining issues with the manuscript that should be addressed prior to

publication:

1. The most significant remaining issue relates to a concern I brought up in the first review - that the MKlp1/Pav knockdown experiments are overinterpreted. Again, although I realize that a statistical test supports the author's conclusion that the MKlp1 spindles are longer, there is so much variability in spindle length in both the control and the MKlp1 depletion conditions that it is difficult to interpret these data, making it likely that any stated changes in spindle length/width are mild at best. Although there are plausible reasons that may explain this variability (the authors note in the rebuttal letter that oocytes in this experiment are harder to stage, that it only represents partial depletion, etc), I still think that this experiment is overinterpreted given the data. The authors noted in the rebuttal letter that they had revised the text to reflect this uncertainty, but multiple strong statements still remain that should be revised:

- Abstract line 11: "MKlp1/Pavarotti is important for shortening the spindle..."
- Page 9, lines 12-14: "...our results suggest that MKlp1/Pav accumulation to the spindle equator plays an important role in shortening and widening of late metaphase spindles..."
- Page 10, lines 20-21: "We showed that MKlp1/Pav is important for shortening and widening..."
- Page 27, lines 22-23: The Figure 5 title is "MKlp1/Pav is important for shortening and widening of the spindle in late oocytes".

These statements (and any others I missed) should all be modified/softened and I would also recommend adding an explicit statement in the results section that the effects observed are mild (along the lines of "although this is a mild effect, potentially due to the fact that the protein is only partially depleted, these data are consistent with the hypothesis that MKlp1 contributes to..." etc.) It also might be helpful if the authors put a statement somewhere (maybe in the Materials and Methods or Figure legend?) explaining the issue with staging the oocytes, so that others don't wonder, as I did, why there is such a large range of spindle lengths represented (if the authors put this in there, I apologize for missing it!). In addition, I would suggest modifying the Figure 5 title, both to re-word the conclusions about MKlp1, but also to reference to the Kinesin-5 data, which is now included in the figure.

2. In adding new data and revising the manuscript, the data in the figures is now in some cases presented out of order, which makes the paper harder for the reader to follow. For example, the data in Figure 3 is now described before the data in Figure 2, and Figure S3 is described before Figure S2. The authors should rearrange the figures to match the flow of the text, making sure that the figures are referenced largely in order.

3. In the abstract (page 2, line 2) and introduction (page 2 lines 25-26), the authors make the general statement that oocytes undergo long arrests. However, this is not true of all species (e.g. *C. elegans*), so these statements should be revised to say "oocytes of most species" or something similar.

4. Page 4, lines 12-13. The authors added a statement to explain that "females were matured in the presence of males and food". A non-fly person will not understand the significance of this statement or why this is important. To make the manuscript accessible to a broader audience, I suggest adding a clause to this sentence (or adding a sentence after it) explaining why this was done and what it means.

5. In the legends for Figures 2, 3, and 5D, the way the n's are represented is confusing. For example, in the Figure 2 legend the authors state "n=20, 10 (Pav),"...etc, but don't say what the two numbers represent. Are these the n's for stage 13 and 14 oocytes, respectively? If so, the authors should

state this, and if not, they should explain what they represent.

6. In the legend for Figure 5, the authors should state what *** denotes.

Minor typos:

- Page 4 line 23: should read "on the other hand, the..."
- Page 5 line 9: should read "localized" (instead of "localize")
- Page 6 line 6: should read "stabilizing spindle bipolarity" (remove "the")
- Page 29, line 9: should read "proteins that do not" (instead of "which")